# Flexing the principal gradient of the cerebral cortex to suit changing semantic task demands

**Zhiyao Gao[1]\*, Li Zheng[2], Katya Krieger-Redwood[1], Ajay Halai[3], Daniel S Margulies[4], Jonathan Smallwood[5], Elizabeth Jefferies[1]\***

[1]Department of Psychology, University of York, New York, United Kingdom; [2]Department of Psychology, University of Arizona, Tucson, United States; [3]MRC Cognition and Brain Sciences Unit, University of Cambridge, Cambridge, United Kingdom; [4]Integrative Neuroscience and Cognition Center (UMR 8002), Centre National de la Recherche Scientifique, Paris, France; [5]Department of Psychology, Queens University, Kingston, Canada

**\*For correspondence:**
zhiyao.gao@york.ac.uk (ZG);
beth.jefferies@york.ac.uk (EJ)

**Competing interest:** The authors declare that no competing interests exist.

**Abstract** Understanding how thought emerges from the topographical structure of the cerebral cortex is a primary goal of cognitive neuroscience. Recent work has revealed a principal gradient of intrinsic connectivity capturing the separation of sensory-motor cortex from transmodal regions of the default mode network (DMN); this is thought to facilitate memory-guided cognition. However, studies have not explored how this dimension of connectivity changes when conceptual retrieval is controlled to suit the context. We used gradient decomposition of informational connectivity in a semantic association task to establish how the similarity in connectivity across brain regions changes during familiar and more original patterns of retrieval. Multivoxel activation patterns at opposite ends of the principal gradient were more divergent when participants retrieved stronger associations; therefore, when long-term semantic information is sufficient for ongoing cognition, regions supporting heteromodal memory are functionally separated from sensory-motor experience. In contrast, when less related concepts were linked, this dimension of connectivity was reduced in strength as semantic control regions separated from the DMN to generate more flexible and original responses. We also observed fewer dimensions within the neural response towards the apex of the principal gradient when strong associations were retrieved, reflecting less complex or varied neural coding across trials and participants. In this way, the principal gradient explains how semantic cognition is organised in the human cerebral cortex: the separation of DMN from sensory-motor systems is a hallmark of the retrieval of strong conceptual links that are culturally shared.

## Editor's evaluation

This work provides important new insights into how semantic association strength influences the function and relationships across brain regions along a topographical structure of cerebral cortex. A principal gradient with the separation of default mode network from sensory-motor systems represents a hallmark of the retrieval of strong conceptual links. This study will be of interest to cognitive neuroscientists, especially those who are interested in semantic cognition.

## Introduction

Recent work has shown that cortical function and intrinsic connectivity change systematically along a 'principal gradient', which has primary sensory and motor cortex at one end, and transmodal regions

implicated in abstract and memory-based functions at the other (*Margulies et al., 2016*). This principal gradient is the dimension of intrinsic connectivity that typically explains the most variance in whole-brain decompositions of human resting-state fMRI data (*Hong et al., 2020*, *Huntenburg et al., 2018*). It is correlated with physical distance along the cortical mantle from primary sensory-motor landmarks. It captures the order of large-scale networks in frontal, temporal, and parietal regions – all of which show transitions from sensory-motor regions, through attention networks to frontoparietal control and then default mode network (DMN) regions. It also captures functional transitions: for instance, opposing ends of the principal gradient are implicated in sensory and memory-based decisions (*Murphy et al., 2019*; *Murphy et al., 2018*; *Lanzoni et al., 2020*). Theories suggest the principal gradient supports increasingly abstract levels of representation, allowing heteromodal regions in the DMN to take on roles that are less directly influenced by externally driven neural activity (*Smallwood et al., 2021*; *Margulies et al., 2016*; *Huntenburg et al., 2018*; *Wang et al., 2020*). In line with this account, the representations of events extracted from movie-watching data are thought to vary in length along this hierarchy, with short events represented in sensory areas and longer events in transmodal regions including within the default mode network (*Baldassano et al., 2017*).

Semantic representation is thought to draw on both ends of this principal gradient (*Huth et al., 2016*; *Barsalou, 2008*; *Kiefer and Pulvermüller, 2012*; *Meteyard et al., 2012*; *Patterson et al., 2007*; *Wang et al., 2010*; *Visser and Lambon Ralph, 2011*; *Murphy et al., 2019*). The graded hub and spoke model proposes that heteromodal concepts are formed at a distance from visual and auditory input regions: diverse sensory processing pathways gradually converge in transmodal areas to form abstract generalisable semantic representations (*Patterson et al., 2007*; *Ralph et al., 2017*). Moreover, we flexibly deploy conceptual information to suit our goals or context: this involves the recruitment of several heteromodal networks, including the DMN, semantic control network (SCN) and multiple-demand network (MDN), which lie at different points along the principal gradient (*Wang et al., 2020*; *Gao et al., 2021*). These networks show differential recruitment across trials depending on the semantic similarity of the items being linked, with higher conceptual overlap associated with stronger activation at the top end of the principal gradient (*Wang et al., 2020*). Anterior temporal lobe (ATL) and angular gyrus (AG), which are allied with DMN, contribute to the relatively automatic retrieval of semantic information strongly supported by long-term memory (*Davey et al., 2015*; *Jefferies et al., 2020*, *Teige et al., 2019*; *Teige et al., 2018*). In contrast, when unusual aspects of knowledge are required – for example, for weak associations – there is greater recruitment of the SCN to shape the meaning that is retrieved (*Davey et al., 2015*; *Jackson, 2021*; *Jefferies, 2013*; *Jefferies et al., 2020*), along with other control regions within the MDN, which responds to higher task demands across domains (*Fedorenko et al., 2013*; *Erb et al., 2013*; *Assem et al., 2020*).

While recruitment is known to vary along the principal gradient according to the demands of the task (*Murphy et al., 2019*; *Wang et al., 2020*), there have been few if any investigations of the way in which neural representation changes over the length of this whole-brain gradient depending on the type of information that is retrieved. Local transitions in the information that is represented neurally have been observed for specific domains. Visual object recognition is supported by a processing hierarchy extending from primary visual cortex, through higher visual regions in inferior temporal cortex, to ventral anterior temporal areas that capture the meaning of concepts across modalities (*Murphy et al., 2017*; *Ju and Bassett, 2020*). A similar functional transition is proposed in medial temporal cortex, which supports detailed, fine-grained spatial representations in posterior hippocampus and more gist or schema like, coarse-grained memory representations in anterior hippocampus (*Poppenk et al., 2013*). Lateral prefrontal cortex has also been proposed to support a rostral–caudal gradient, representing specific motor actions in posterior regions through to more abstract goals in anterior regions (*Badre and D'Esposito, 2009*; *Badre and Nee, 2018*). However, these studies have not examined changes in connectivity gradients as a function of task demands. Dimensionality reduction techniques like principal component analysis (PCA) can also quantify the number of components needed to explain a specific amount of variance in the neural response of a given brain region, providing an index of the complexity or dimensionality of the representational space over time (*Ahlheim and Love, 2018*). Neural representation in unimodal regions is thought to have many dimensions, allowing the cortex to discriminate between similar inputs, while the representational space may have fewer dimensions in transmodal DMN regions, allowing a variety of inputs to be encoded into common activity patterns that generalise knowledge across circumstances

(*Ju and Bassett, 2020*). For example, we can easily tell the difference between 'Dalmatian' and 'Bulldog' yet understand their abstract similarity as 'dogs'. Category learning based on an abstract rule is also associated with a prefrontal response characterised by fewer dimensions (*Mack et al., 2020*).

In the current study, we manipulated the degree to which ongoing semantic cognition was aligned with long-term semantic knowledge and quantified the similarity of the multivariate response to each trial along the principal gradient. Our task asked participants to generate links between pairs of words that varied in their associative strength, including strongly associated items which often co-occur, weakly linked items and unrelated pairs. We asked whether functional connectivity patterns during this task would mirror the principal gradient and vary flexibly according to the associative strength of the items in each trial. We also asked whether neural representations along the principal gradient differed in terms of their complexity. With less support from long-term memory, unusual patterns of conceptual retrieval might be supported by control regions that promote specific sensory-motor features, reducing the functional separation of the two ends of the principal gradient in terms of their patterns of connectivity. In contrast, frequently occurring patterns of conceptual retrieval may emerge from long-term memory in the absence of additional constraints from brain regions; this retrieval state might be associated with increased functional separation between DMN and sensory-motor regions, allowing memory to underpin cognition irrespective of ongoing sensory-motor experience. We might also expect the dimensionality of neural representations in a semantic task to decrease from unimodal to transmodal areas along the principal gradient, reflecting increasingly abstract and culturally shared representations towards the apex of the gradient, particularly when participants retrieve strong associations that link words together in specific well-established ways.

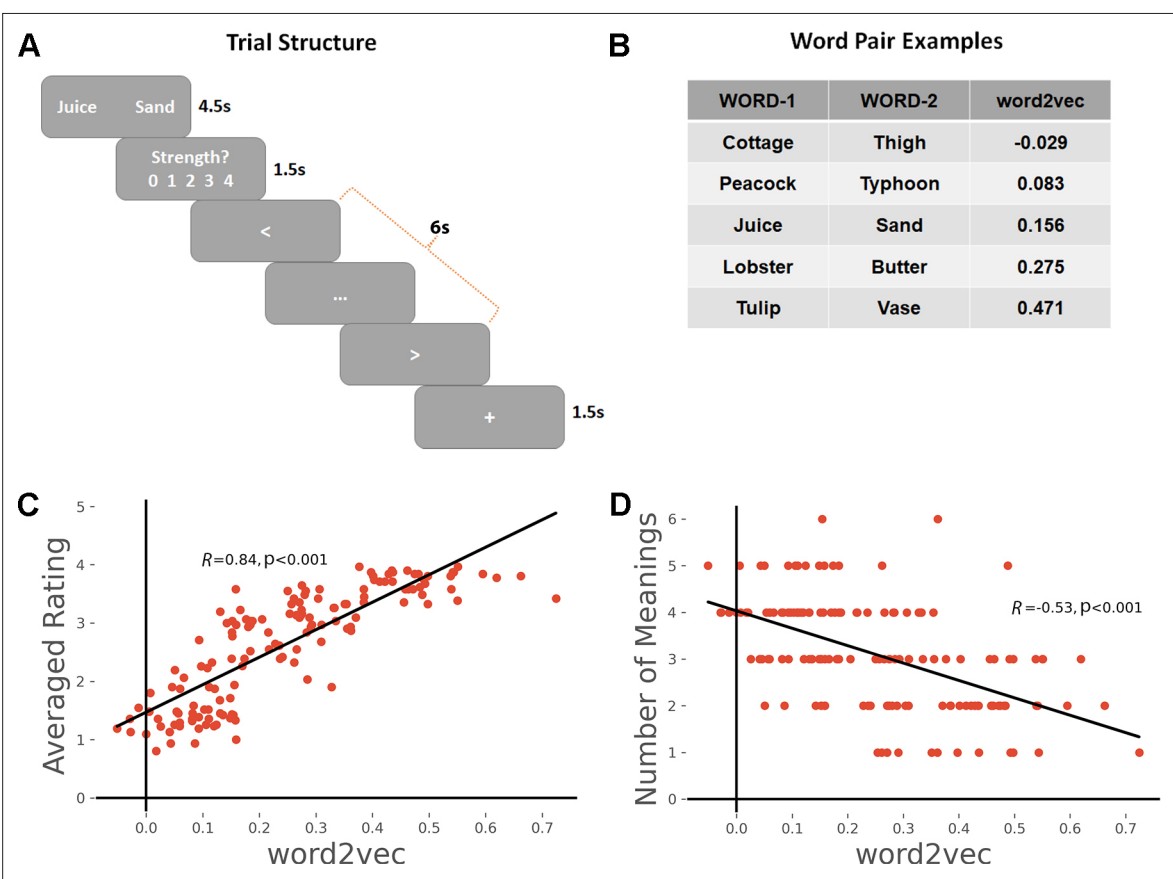

**Figure 1.** (**A**) Task schematic; participants were asked to produce a link between the words and rate the strength of the link. (**B**) Word-pair examples with association strength ranging from weak to strong, based on word2vec scores. (**C**) The average rating of link strength for each word-pair was positively associated with its word2vec score. (**D**) The number of meanings produced by the group of participants for each word-pair was negatively associated with the word2vec score. All data reflect $n = 31$ independent participants.

# Results

## Behavioural results

Participants' in-scanner ratings of the strength of the link they made between the words in each pair were strongly correlated with word2vec values as expected (Pearson $r = 0.84$, p < 0.001), see *Figure 1C*. We also predicted that the links formed by participants would be more consistent across individuals for word-pairs that were more strongly associated, as these links could be generated from dominant aspects of long-term semantic memory, which should be shared across individuals. To test this hypothesis, we collected the subjective reports of the links that participants formed on each trial and counted how many unique meanings/links were produced. There was a significant negative correlation between word2vec and the number of meanings generated ($r = -0.53$, p < 0.001), confirming that participants generated more variable patterns of semantic retrieval for weakly associated word-pairs, see *Figure 1D*.

On the chevron task, the average reaction time was 0.336 s and the accuracy was 72.6%; these data suggest the task was challenging enough to engage the participant but not so difficult that they could not do it. The fast pace of this task reduced participants' ability to think about semantic links across trials.

## fMRI results

### The principal gradient of cortical organisation was modulated by strength of association

fMRI was used to assess how the topographical organisation of the neural response to the task changed as a function of strength of association. Recent research decomposing whole-brain intrinsic connectivity patterns in resting-state fMRI into their spatial components has revealed a 'principal gradient' of intrinsic functional connectivity, which captures the separation between transmodal memory systems and unimodal sensory-motor cortex at rest (*Margulies et al., 2016*). Here, we decomposed informational connectivity matrices in the same way.

We first verified that the principal gradient of cortical organisation could be identified from task-based informational connectivity matrices as well as from intrinsic patterns of connectivity at rest, and that this dimension of cortical organisation was robust across different cognitive tasks. PCA was used to decompose the group-averaged connectivity matrices, with the resulting components (or gradients) reflecting spatial patterns of connectivity across the cortex. These components were ordered by the variance they explained in the initial connectivity matrix. Within each gradient, similar values reflected similarity in patterns of connectivity. The decomposition was performed separately for the semantic task (TR 4–6) and the chevron task (TR 9 and 10). This similarity between cortical gradients was established by using spin permutation tests, which preserve the spatial autocorrelation present within gradients. Statistical significance was determined by comparing with a null distribution using the permutation approach (5000 permutations), in which we randomly rotated the spatial patterns and measured the correlations between maps that preserve spatial autocorrelation (*Alexander-Bloch et al., 2018*). The first gradient, accounting for the greatest variance in informational connectivity, was highly overlapping with the principal gradient of intrinsic connectivity identified by prior work (*Margulies et al., 2016*), for both the semantic and chevron tasks (Spin permutation p value: $p_{spin} < 0.001$), see *Figure 2—figure supplement 1*. The gradient which explained the second most variance for the semantic task was also positively correlated with the corresponding gradient derived from resting data ($p_{spin} < 0.001$), demonstrating a pattern of distinguishing visual areas from auditory/motor areas. The gradient which explained third most variance for the semantic task was only positively correlated with the corresponding gradient derived from resting data in the left hemisphere ($p_{spin} < 0.001$), while a negative relationship was found for the right hemisphere, see *Figure 2—figure supplement 2*.

Next, we examined our core hypothesis that the principal gradient of cortical organisation is flexed according to the demands of cognitive tasks. Since our behavioural data highlight how more strongly associated word-pairs elicit more similar patterns of semantic retrieval across participants, the neurocognitive response in these trials is expected to reflect the retrieval of readily accessible semantic information from long-term memory, which is shared across people. This retrieval state is assumed to be driven by relatively abstract conceptual representations in transmodal regions as opposed to sensory-motor features that vary across experiences; consequently, the retrieval of strong associations

might strengthen the principal gradient (i.e. it might drive functional separation between transmodal and unimodal cortex), relative to the more idiosyncratic and controlled patterns of retrieval required for weaker associations and unrelated words.

We derived estimates of the principal gradient of informational connectivity for strong and weak associations separately, dividing trials according to their word2vec values and focussing on the semantic response in TR 4–6. This analysis revealed a significant increase in principal gradient values for strong versus weak associations in transmodal areas implicated in semantic cognition, including in bilateral prefrontal cortex, temporal lobe, and presupplementary motor cortex. These regions, shown in warm colours on *Figure 2B*, are found towards the DMN apex of the principal gradient. The reverse of this pattern (i.e. lower principal gradient values for strong versus weak associations) was found in sensory-motor regions towards the bottom of the principal gradient. These regions are shown in cool colours on *Figure 2A*. The principal gradient difference between middle and weak association trials of the semantic task showed a similar pattern, see *Figure 2—figure supplement 3*.

In a control analysis, we also examined the chevron task period (TR 9 and 10) for the same subsets of trials (i.e. divided on the basis of word2vec values for the immediately preceding semantic decision). A different pattern was found: although there were still differences between these sets of trials that might have reflected the consequences of earlier variation in cognitive effort or retrieval, these differences were substantially weaker for the chevron than the semantic period in transmodal regions (see *Figure 2C*). In addition, these differences were not systematically aligned with the principal gradient: some DMN regions showed higher gradient values for chevron decisions following strong versus weak associations (e.g. in right lateral temporal and right medial prefrontal cortex), while other DMN regions showed the reverse pattern (e.g. in left AG). In addition, visual and motor cortex did not generally show lower principal gradient values for chevron trials following strong versus weak associations; these regions often showed opposing effects for the chevron and semantic TRs, see *Figure 2—figure supplement 4*.

To examine how the principal gradient was modulated by the tasks within pre-defined functional networks that are relevant to semantic representation and control demands, we conducted a two-way repeated-measures analysis of variance examining the within-participant factor of task (semantic vs. chevron) in different brain networks. These networks corresponded to (1) regions implicated in semantic cognition that fell outside a meta-analytic map of semantic control (*Jackson, 2021*) – focused on anterior temporal cortex and AG; (2) SCN regions showing activation across a range of manipulations of semantic task difficulty (*Jackson, 2021*) yet which are outside the MDN (*Fedorenko et al., 2013*) – including left anterior inferior frontal gyrus and posterior middle temporal gyrus; (3) regions falling in both SCN and MDN (SCN + MDN) – including inferior frontal sulcus and pre-supplementary motor area, and (4) MDN regions not implicated in semantic cognition; see methods. We found a significant interaction between brain network and task ($F(2.53, 75.887) = 8.345$, $p < 0.001$, $\eta_p^2 = 0.218$) and significant main effects of task ($F(1, 30) = 21.947$, $p < 0.001$, $\eta_p^2 = 0.414$) and network ($F(2.01, 60.288) = 3.338$, $p = 0.042$, $\eta_p^2 = 0.100$), see *Figure 2D*. Simple *t*-tests revealed significant differences in principal gradient values for the semantic task across SCN and SCN + MDN compared with MDN regions ($t(30) = 3.259$, $p = 0.0167$, 95% confidence interval [CI] = [0.279 ~ 1.215], Cohen's $d = 0.827$; and $t(30) = 3.904$, $p = 0.003$, 95% CI = [0.334 ~ 1.066], Cohen's $d = 0.965$, respectively), while no significant differences between networks were found for the chevron task ($p > 0.682$). Bonferroni correction was applied. These results show that networks implicated in semantic representation and control, but not executive control across domains, have lower gradient values when weaker associations are retrieved, suggesting less separation in connectivity from sensory-motor cortex.

To quantify the semantic task principal gradient difference between strong and weak associations at a finer-grained scale, we used Schaefer's 400-region whole-brain parcellation. We assessed the correlation between the gradient difference value for each parcel (between strong and weak associations) and resting-state gradient reported by *Margulies et al., 2016*. There was a positive correlation for the semantic task: parcels higher on the resting-state gradient tended to show a larger effect of associative strength (strong > weak) on task-derived gradient values ($r = 0.369$, $p < 0.001$), while no correlation was found in the chevron task ($r = -0.038$, $p = 0.45$), see *Figure 2E, F*. These results confirmed that variation in the strength of association between words influenced task-induced gradients of informational connectivity, with brain regions at the heteromodal end of the principal gradient showing more similar connectivity patterns to each other, and more separation from unimodal regions,

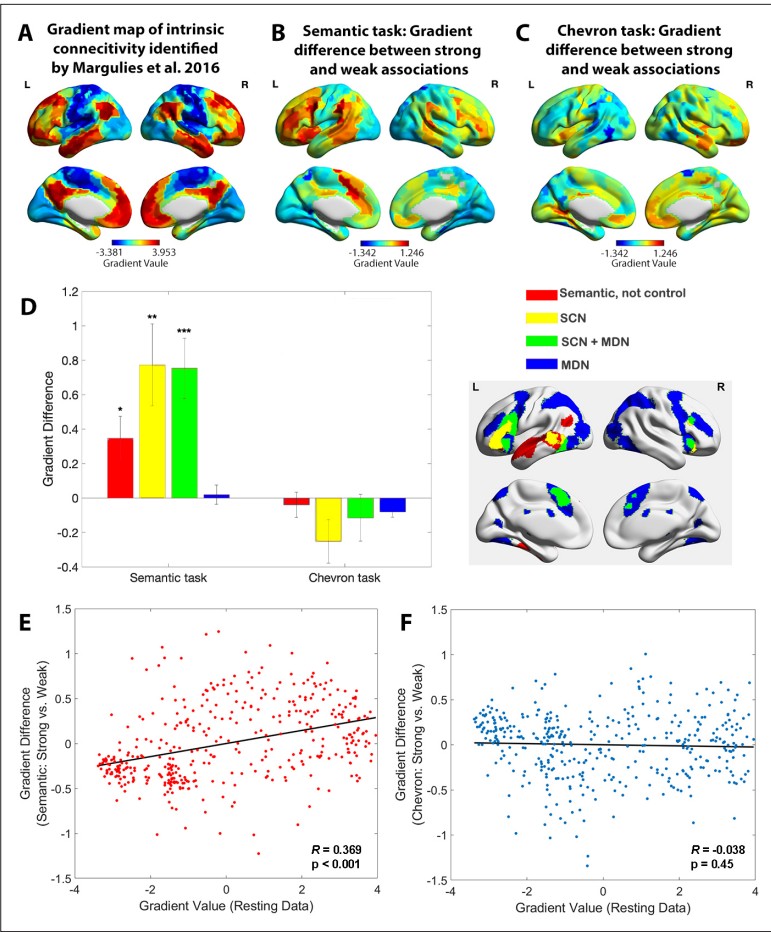

**Figure 2.** The principal gradient of cortical organisation was modulated by strength of association. (**A**) The principal gradient map identified from intrinsic connectivity reported by *Margulies et al., 2016*. (**B**) The difference map of principal gradient values during the semantic task comparing strong and weak associations estimated using the informational connectivity approach and brain responses in TR 4–6. (**C**) The difference map of principal gradient values during the chevron task, comparing time points that followed strong and weak associations, estimated using the informational connectivity approach and brain responses in TR 9–10. (**D**) Left panel. The difference in principal gradient values between strong and weak association trials showed a significant interaction effect between functional networks and tasks ($F(2.53, 75.887) = 8.345$, $p < 0.001$, $\eta^2_p = 0.218$). Error bars represent the standard error of gradient difference across parcels within each network. Right panel. Functional networks: (i) semantic not control, (ii) within the semantic control network (SCN) but outside multiple-demand cortex (DMN), (iii) within both SCN and multiple-demand network (MDN), and (iv) falling in MDN regions not implicated in semantic cognition. (**E**) Using Schaefer's 400 parcellation, gradient difference values in each parcel for strong and weak associations in the semantic task were positively associated with the principal gradient values of these parcels, estimated from independent resting data. (**F**) No correlation between informational connectivity differences per parcel and principal gradient values estimated from resting-state fMRI data was observed for the chevron task when comparing timepoints that followed strong and weak semantic associations. *$p < 0.05$, **$p < 0.01$, ***$p < 0.001$. All data reflect $n = 31$ independent participants.

The online version of this article includes the following figure supplement(s) for figure 2:

**Figure supplement 1.** The principal gradient identified by informational connectivity using semantic neural responses and chevron neural responses (upper panel).

**Figure supplement 2.** Top 3 gradients which explained most variance derived from resting connectivity data reported by *Margulies et al., 2016* (left panel) and from informational connectivity data from a semantic task (right panel).

**Figure supplement 3.** The principal gradient difference between middle 1/3 and bottom 1/3 trials for the semantic neural responses (**A**) and for chevron task signals (**B**).

*Figure 2 continued on next page*

*Figure 2 continued*

**Figure supplement 4.** Gradient 1 differences between strong and weak associations in *Yeo et al., 2011* 7 networks for semantic and chevron task separately.

**Figure supplement 5.** The principal gradient estimated by using traditional functional connectivity (Pearson correlation of averaged time series between any pair of ROIs) was insensitive to the association strength in the semantic task.

during semantic decisions involving strongly linked items. These results were confirmed using spin permutation to control for the effects of spatial autocorrelation: again, there was a significant correlation between the resting-state gradient and the difference of gradient values between strong and weak associations for the semantic response ($p_{spin} < 0.001$), while no such effect was found for the chevron task ($p_{spin} = 0.195$).

## Representational space changed along the principal gradient

The analysis above suggests that the functional role of cortical parcels in semantic cognition depends on their location on the principal gradient. This gradient captures a functional hierarchy of processing that extends from sensory-motor features to more abstract long-term memory representations (*Wang et al., 2020*; *Smallwood et al., 2021*; *Margulies et al., 2016*; *Huntenburg et al., 2018*), echoing the way in which semantic cognition draws on representations from sensory-motor features to abstract and heteromodal conceptual information in long-term memory (*Ralph et al., 2017*; *Patterson et al., 2007*). Given that strong associations are associated with greater separation of sensory-motor features from memory-based cognition along the gradient, we might also expect regions at the apex of the gradient to reflect simpler representational states when strong associations are being retrieved. This is because the long-term semantic knowledge underpinning strong associations is highly predictable, abstract and shared across participants.

We used the dimensionality or heterogeneity of the voxels' responses in each parcel to index the complexity of neural representation. We applied a searchlight-based PCA approach to the neuroimaging data for each trial, concatenated across participants. By calculating the number of components which together explained more than 90% of the variance in each local cube, we generated a neural dimensionality map for each trial. *Figure 3A* shows this dimensionality map averaged across all trials. Some heteromodal regions at the top of the principal gradient overlapping with DMN showed activity characterised by many dimensions (e.g. lateral and medial temporal cortex), while other regions with similar gradient values had fewer dimensions (inferior parietal and precuneus cortex).

To quantify the degree to which the dimensionality followed the resting-state principal gradient, we correlated dimensionality estimates with principal gradient values derived from intrinsic connectivity data by *Margulies et al., 2016*. We also assessed whether this relationship was influenced by associative strength, averaging estimates for sets of four trials with similar word2vec values using a 'sliding window' approach. The results showed a weak negative correlation overall (when all trials were included) between dimensionality and principal gradient values ($r = -0.155$, $p = 0.0018$), suggesting that the dimensionality of neural coding decreased, at least to some extent, along the principal gradient from unimodal to heteromodal regions. Importantly, the magnitude of this negative correlation increased as a function of association strength ($r = -0.425$, $p < 0.001$, see *Figure 3D*): more strongly associated word-pairs generated neural responses at the heteromodal end of the gradient which had even lower dimensionality. This pattern remained significant even when regions in limbic network were removed due to their low tSNR ($r = -0.346$, $p = 0.038$). Additional control analyses assessed the dimensionality in the chevron task and its relation to the strength of association in the preceding semantic trial. There was a weak negative corelation between overall dimensionality (averaged across all trials) and the principal gradient estimated by resting-state data ($r = -0.08$, $p < 0.001$), in line with the expectation of more abstract representations at the top of the gradient. The relationship between the resting-state principal gradient and dimensionality (estimated for sets of four trials with similar strength of association) did not vary with the association strength of the preceding semantic trial ($r = -0.18$, $p = 0.28$, see *Figure 3—figure supplement 1*).

A direct comparison of weak and strong associations (contrasting the bottom and top third of the trials arranged according to associative strength) revealed that the neural response had significantly higher dimensionality for weakly related trials in regions implicated in semantic control, including

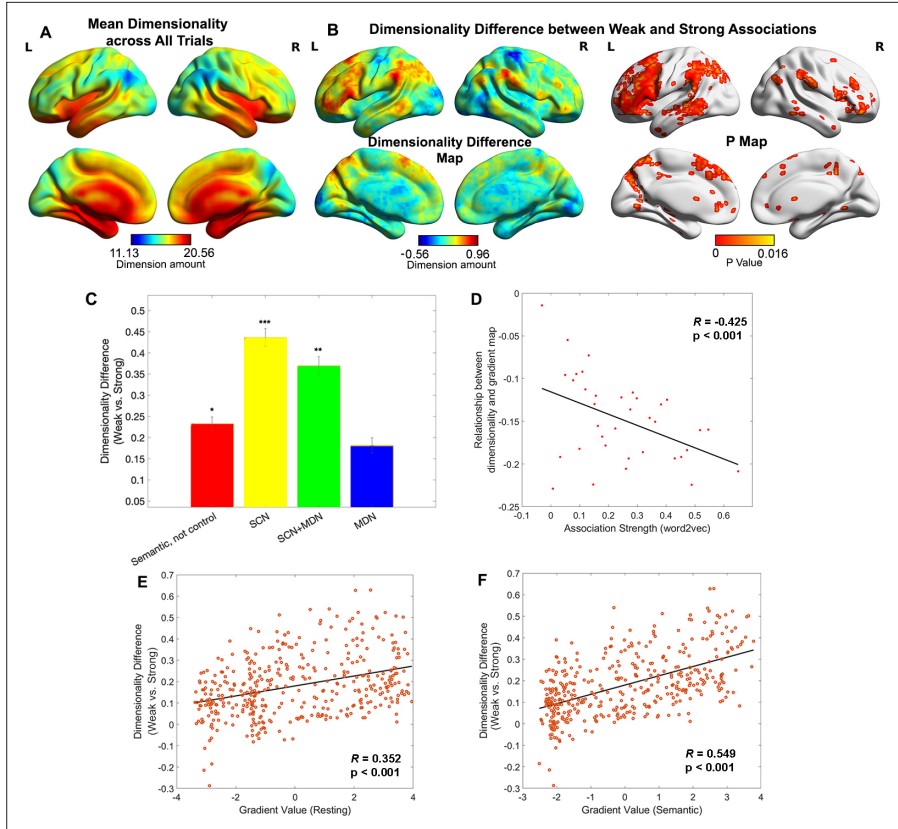

**Figure 3.** Representational space changed along the principal gradient. (**A**) Dimensionality map averaged across all trials. (**B**) Dimensionality difference between weak and strong associations (left panel) and the corresponding p map (right panel), with FDR correction applied, $q = 0.05$. (**C**) Dimensionality difference between weak and strong in functional networks implicated in semantic representation and control demands (Semantic, not control: p = 0.017, semantic control network (SCN): p < 0.0001, SCN + multiple-demand network (MDN): p = 0.002, MDN: p = 0.245, Wilcoxon rank sum test was conducted and p values were adjusted using Bonferroni correction). Error bars represent the standard error of dimensionality differences across parcels within each network. (**D**) The relationship between dimensionality and the principal gradient was generally negative (all data points are below 0) and negatively correlated with the associative strength between word-pairs. We averaged the dimensionality map for every four trials arranged in order from weak to strong according to their word2vec values for this analysis. (**E**, **F**) The dimensionality difference between weak and strong trials was positively correlated with the location of each parcel on the principal gradient estimated from resting-state fMRI (in E) and informational connectivity during the semantic task (in F). Each data point represents a parcel from the Schaffer 400 parcellation. Note: the dimensionality difference was examined by subtracting strong from weak associations; this contrast is in the opposite to the gradient difference analysis, since we anticipate higher gradient values and fewer neural dimensions for strong associations in similar brain regions. *p < 0.05, **p < 0.01, ***p < 0.001. All data reflect $n = 31$ independent participants.

The online version of this article includes the following figure supplement(s) for figure 3:

**Figure supplement 1.** We averaged the dimensionality map in the chevron task for every four trials arranged in order of the word2vec values for the preceding semantic task from weak to strong for this analysis.

**Figure supplement 2.** Dimensionality differences between weak and strong association trials in classical resting networks (right) (*Yeo et al., 2011*).

**Figure supplement 3.** The significance p map of dimensionality differences between weak and strong associations in which the dimensions of the activation pattern were determined using different criteria (explaining 60% [left panel] or 75% [right panel] of the variance), with FDR correction applied, $q = 0.05$.

inferior frontal gyrus, presupplementary motor cortex, and posterior middle temporal gyrus in the left hemisphere, as well as in regions of the default mode network including the bilateral AG, precuneus, medial prefrontal cortex, and left anterior temporal pole (*Figure 3B*). No clusters showed the reverse pattern (i.e. higher dimensionality for activation patterns during the retrieval of strong associations). Confirmatory analyses examined the effect of association strength on dimensionality in networks defined using resting-state data (*Yeo et al., 2011*) and functional recruitment during semantic and control tasks. This confirmed the strongest dimensionality difference was within semantic control areas, with no effect of association strength in multiple-demand areas, see *Figure 3C* and *Figure 3—figure supplement 2*. To check the robustness of our results, we performed a series of control analyses in which the dimensions of the activation pattern were determined using different criteria (explaining 60% or 75% of the variance) and obtained highly similar results, see *Figure 3—figure supplement 3*. Using Schaefer's 400-region parcellation, we established a significant positive correlation between the dimensionality change across weak and strong associations and the principal gradient value for each parcel obtained from both intrinsic connectivity ($r = 0.352$, $p < 0.001$, *Figure 3E*) and the informational connectivity of our semantic task ($r = 0.549$, $p < 0.001$, *Figure 3F*). These results show that the semantic representational space is more strongly modulated by association strength towards the heteromodal end of the principal gradient, implicating these regions in individual differences in semantic cognition.

## Relationships between the principal gradient, representational dimensionality, and semantic cognition

To identify regions in which neural activity patterns could predict differences in semantic cognition between participants, we performed a second-order RSA to examine the semantic-brain alignment using a whole-brain searchlight approach. A semantic similarity matrix, based on the correlation of participants' ratings of associative strength across trials (reflecting the global similarity of neural states of retrieval between participants; left-hand panel of *Figure 4A*), was positively associated with neural pattern similarity in inferior frontal gyrus, posterior middle temporal gyrus, right ATL, bilateral lateral and medial parietal cortex, pre-supplementary motor area, and middle and superior frontal cortex (right-hand panel of *Figure 4A*). Functional masks defined by semantic and control tasks and classical resting-state networks (*Yeo et al., 2011*) confirmed that semantic control regions and DMN showed the strongest association with individual differences in associative strength ratings across participants, see *Figure 4—figure supplement 1*. Therefore, heteromodal regions implicated in semantic representation (within the default mode network) and semantic control were associated with individual differences in semantic cognition across participants.

Next, we examined whether the strength of the semantic-brain alignment could be predicted by either dimensionality differences between strongly associated and weakly associated trials or the semantic task principal gradient. Using Schaefer's 400-region parcellation, we assessed the correlation between these three variables across parcels. Semantic-brain alignment was positively associated with dimensionality differences between strongly associated and weakly associated trials ($r = 0.506$, $p < 0.001$, *Figure 4C*), and with the principal gradient derived from the semantic task ($r = 0.42$, $p < 0.001$, *Figure 4B*) and from intrinsic connectivity ($r = 0.255$, $p < 0.001$; see *Figure 4—figure supplement 2*). These analyses show that individual differences in semantic cognition were associated with variation in the neural response more strongly in parcels showing less variable or complex neural responses to strongly associated items, and towards the DMN apex of the principal gradient.

Given that the principal gradient correlated with both differences in the dimensionality of neural responses between strong and weaker associations and semantic-brain alignment, we reasoned that the hierarchy of brain organisation captured by the principal gradient might influence semantic cognition via effects on neural dimensionality. To test this hypothesis, we performed a mediation analysis in which the principal gradient was the predictor, semantic-brain alignment was the dependent variable, and the effect of strength of association on the dimensionality of the neural response was the mediator. The significance of the mediation effect was tested by using the bootstrapping approach with 5000 samples. Consistent with our prediction, we found the correlation between the principal gradient and semantic-brain alignment could be partially explained by this dimensionality difference (indirect effect: $a \times b = 0.0053$, 95% CI = [0.0039–0.01], $p < 0.001$, explaining 51.7% variance; direct effect: $c' = 0.0049$, 95% CI = [0.0026–0.01], $p < 0.001$). This result suggests the apex of the principal

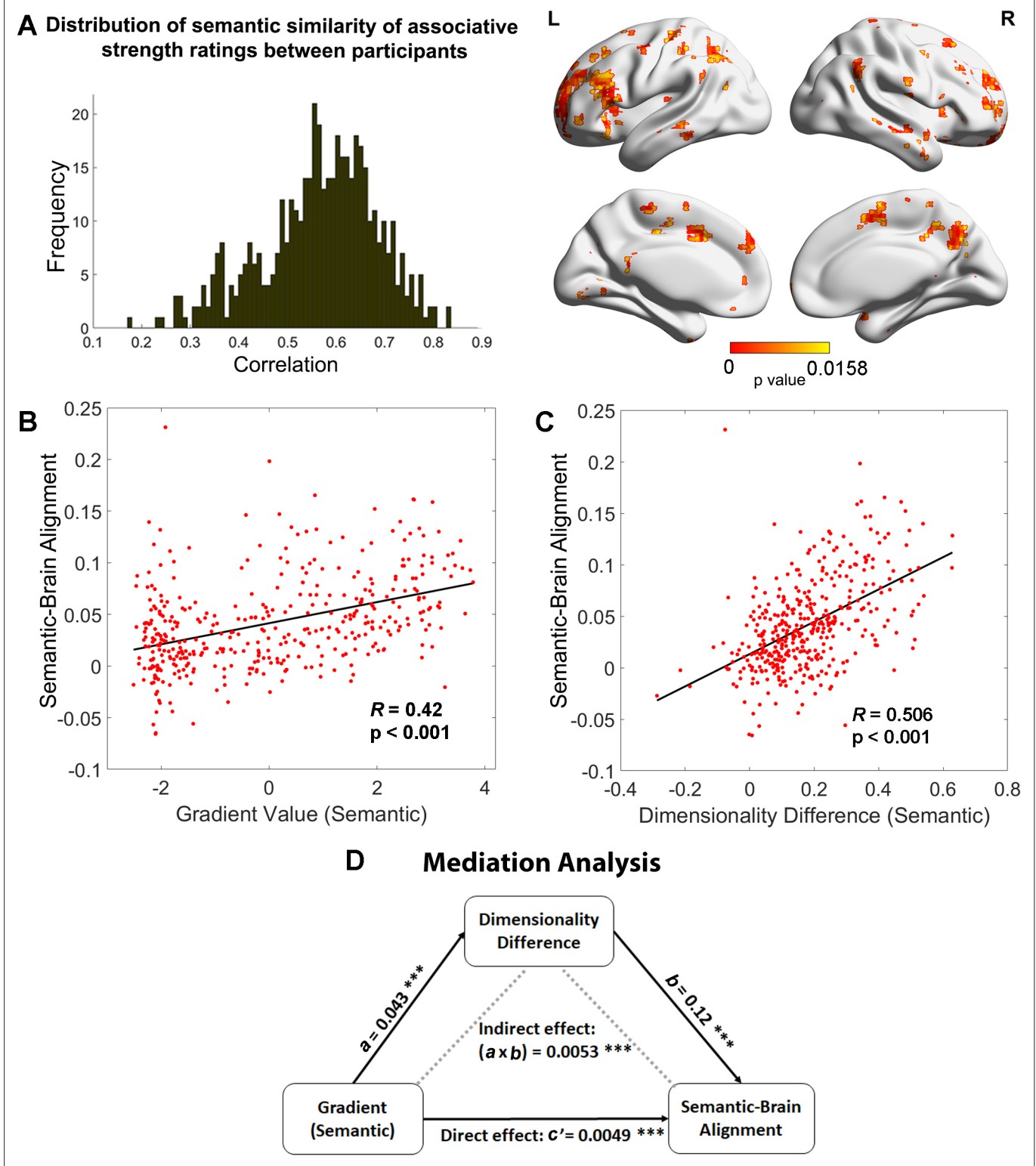

**Figure 4.** Relationships between the principal gradient, representational dimensionality, and semantic cognition. (**A**) Left: The distribution of similarity in associative strength ratings between participants. We measured the semantic similarity by correlating the rating patterns of associative strength between any pair of participants (x-axis: correlation) and plotted the distribution of semantic similarity (y-axis: frequency). Right: More similar neural patterns between participants and trials were positively associated with similarity in semantic ratings; FDR correction was applied, *q* = 0.05. (**B**) The semantic task principal gradient was positively associated with semantic-brain alignment shown in the right side of (**A**); each point represents one parcel. (**C**) The difference in the dimensionality of the neural response between weak and strong associations (weak–strong) was also positively associated with

*Figure 4 continued on next page*

*Figure 4 continued*

the semantic-brain alignment estimated from *Figure 4*; each point represents one parcel. (**D**) Mediation analysis. The semantic task principal gradient was the independent variable, the dimensionality difference between weak and strong associations was the moderator and semantic-brain alignment was the dependent variable. This analysis revealed that the relationship between the semantic task principal gradient and semantic-brain alignment could be partially explained by the difference in dimensionality of the neural representational space between strong and weak trials (explained 51.7% variance), ps < 0.001. ***p < 0.001. All data reflect *n* = 31 independent participants.

The online version of this article includes the following figure supplement(s) for figure 4:

**Figure supplement 1.** The semantic-brain alignment estimated using second-order RSA in functional networks implicated in semantic representation and control demands (left) or in resting networks (right) (*Yeo et al., 2011*).

**Figure supplement 2.** The principal gradient derived from resting data (1) was positively associated with semantic-brain alignment (*r* = 0.255, p < 0.001).

gradient might relate to individual differences in semantic cognition because its behaviour reflects the extent to which information is common across individuals: when there is greater semantic-brain alignment at the top of the gradient, the neural patterns for strong associations also show less variation and complexity than for weak associations (*Figure 4D*).

## Discussion

This study modulated the associative strength between word-pairs to delineate the neural patterns underpinning semantic retrieval along the principal gradient of cortical organisation. We combined dimensionality reduction methods (gradient analysis) with informational connectivity capturing the multivariate similarity between brain regions during the retrieval of semantic associations. This analysis showed that the presentation of strong associations increased the functional separation between unimodal and transmodal regions at opposite ends of the principal gradient. Stronger associations elicited higher gradient values in transmodal regions and lower gradient values in unimodal regions, suggesting that commonly occurring patterns of cognition mediated by long-term memory involved greater informational separation from sensory-motor features. In contrast, more idiosyncratic patterns of retrieval involved less separation between transmodal and unimodal representations, perhaps because participants fell back on specific features to generate a link when dominant aspects of long-term memory were insufficient for the task. This change was greatest in semantic control regions, which were close to the apex of the principal gradient along with DMN at rest and during the retrieval of strong associations, yet showed substantially lower gradient values during the retrieval of weak associations. We also found a larger decrease in the dimensionality of neural representations from unimodal to transmodal regions along the principal gradient when strong associations were retrieved; the response towards the DMN apex of the gradient was less complex or variable across participants for strong associations, reflecting the way that these trials relied on aspects of knowledge or retrieval states that could generalise across the specific experiences of individuals. The neural representation of weakly associated word-pairs was associated with higher dimensionality or heterogeneity in distributed transmodal areas, including in lateral DMN and semantic control regions. Finally, mediation analysis showed that the relevance of cortical gradients to individual differences in rated semantic similarity was partially explained by neural dimensionality differences between weak and strong associations. Therefore, the neural organisation of semantic cognition not only reflects global patterns of brain connectivity but changes in global connectivity patterns associated with variation in transmodal neural representation.

Notably, while most gradient studies generate connectivity matrices by correlating mean univariate signals between ROIs over time, normally using resting-state fMRI data (*Huntenburg et al., 2018*; *Margulies et al., 2016*), we employed a novel informational connectivity approach to measure the similarity of multivariate response patterns between ROIs during a task (*Anzellotti and Coutanche, 2018*; *Coutanche and Thompson-Schill, 2013*). The cortical gradient explaining the most variance in informational connectivity was highly correlated with the principal gradient estimated from resting data (*Margulies et al., 2016*) using the traditional univariate approach. [This relationship between task-based informational connectivity and resting-state fMRI analysis was also found for the gradient which explained the second most variance for the semantic task. The gradient which explained the third most variance showed a moderate correlation with the third gradient derived from resting-state

data in the left hemisphere. However, the correlation was negative in the right hemisphere, suggesting an influence of the task: semantic cognition might be expected to separate the connectivity patterns of left and right hemispheres, since it is left lateralised.] Moreover, the variance explained by the principal gradient was higher for the decomposition of informational connectivity than for time-series correlation (40–50% vs. 25%). Therefore, the current study helps to validate a new approach to deriving cortical gradients from task data which was sensitive to the effects of associative strength in semantic cognition.

The human capacity to retrieve abstract meaning from sounds, visual objects, words, and phrases exceeds the capacities of other animal species; however, the development of this ability is poorly understood (*Enge et al., 2021*). One recent study *Dong et al., 2021* found that, in children, the overarching organisational gradient is anchored within the unimodal cortex, between somatosensory/motor and visual territories, while in adolescence, the principal gradient of connectivity transitions into an adult-like spatial framework, with the default network at the opposite end of a spectrum from primary sensory and motor regions. Whether the development of abstract semantic representations is related to this gradual emergence of the principal gradient in the brain remains an open question which can be addressed in future studies.

Our findings are broadly consistent with the controlled semantic cognition framework (*Jefferies et al., 2020*, *Ralph et al., 2017*), which proposes that heteromodal semantic cognition emerges from the distillation of sensory-motor features into more abstract concepts; semantic control processes then shape the responses across this network to suit the demands of the task. There is strong overlap between the semantic network recruited during tasks and DMN regions identified from the parcellation of resting-state data, for example in inferior and lateral temporal and lateral parietal cortices (*Davey et al., 2016*; *Jackson et al., 2016*; *Seghier et al., 2010*; *Humphreys and Lambon Ralph, 2015*). When semantic retrieval is focused on dominant features of conceptual knowledge, automatic retrieval might emerge from the unconstrained behaviour of these semantic 'hub' regions, including ATL and potentially other sites with similar patterns of connectivity, such as left AG (*Humphreys and Lambon Ralph, 2015*). Gradient analysis of both task and rest data shows that these transmodal regions have a high degree of pattern similarity to each other, and weaker similarity to sensory-motor cortex. Long-term knowledge amplifies the informational distance between the opposite ends of the principal gradient, suggesting that when strong associations are retrieved, areas associated with abstract and heteromodal conceptual representation separate from specific sensory-motor features and attentional networks at the opposite end of the gradient, as these elements are not needed for task-relevant patterns of retrieval. In contrast, when retrieval is focused on unusual aspects of semantic knowledge, semantic control processes shape retrieval to suit the demands of the task – for example, inhibiting dominant associative links and retrieving specific features that support specific and unusual connections between words (*Jackson, 2021*; *Jefferies et al., 2020*, *Gao et al., 2021*, *Noonan et al., 2013*). In these circumstances, stronger activation is seen within the 'semantic control network' comprising inferior frontal gyrus, posterior middle temporal gyrus, and pre-supplementary cortex (*Jackson, 2021*; *Noonan et al., 2013*, *Gao et al., 2021*).

We found a positive linear relationship between the difference of gradient values across weak and strong associations and principal gradient location, providing evidence that regions towards the DMN apex can flex their patterns of connectivity depending on the demands of the task. Semantic control regions showed the largest changes in gradient values and dimensionality between strong and weak associations, suggesting that these regions separate from other heteromodal areas towards the DMN apex of the gradient and interact more strongly with attentional networks and unimodal regions to generate unusual connections between words. While our results are in line with the controlled semantic cognition framework in general, multiple cognitive processes might contribute to the differences between stronger and weaker associations in our task, including the requirement to generate a novel link, as opposed to recognising an existing link. There are also likely to be differing demands on more automatic versus more controlled retrieval, imagery and processes associated with creativity.

Our gradient analyses are highly consistent with inter-voxel similarity analysis across participants and trials. Using this approach to index the complexity of neural representations, researchers have revealed a graded transition from coarse to fine-grained representations along the long axis of the hippocampus from posterior to anterior regions (*Brunec et al., 2018*). Lower inter-voxel similarity over time in the hippocampus has been linked to more differentiated memory representations (*Chanales*

et al., 2017). Motivated by these studies, we measured the heterogeneity of the voxels' responses in local areas to assess the dimensionality of representational space during semantic retrieval. We found a dimensionality decrease from unimodal to transmodal regions along the principal gradient which was larger for strong associations. This finding is consistent with recent studies showing that multivariate patterns within transmodal areas of ventral ATL represent meaning irrespective of presentation modality, feature input, or language (*Murphy et al., 2017*; *Coutanche and Thompson-Schill, 2015*; *Correia et al., 2014*) – this abstraction of conceptual representations away from surface features of the input is likely to be critical for the generalisation of knowledge across modalities and diverse contexts and to support shared understanding across individuals. In line with this view, higher-dimensional spaces are thought to enable the discrimination of similar inputs but consume more resources to embed the information, while lower-dimensional spaces facilitate generalisation and novel learning across stimuli or situations (*Badre et al., 2021*; *Ju and Bassett, 2020*). DMN regions near the apex of the principal gradient may support more abstract representations of meaning when the retrieval demands of a task are well aligned with the information stored in long-term semantic memory.

Consistent with this hypothesis, neural representations showed higher dimensionality for weak associations in transmodal regions, including the semantic 'hub' regions in ATL and AG, and semantic control regions in lateral and medial prefrontal, parietal, and temporal cortex (*Humphreys and Lambon Ralph, 2015*, *Jackson et al., 2016*; *Jefferies, 2013*; *Jefferies et al., 2020*; *Seghier et al., 2010*). Notably, the dimensionality difference peaked in semantic control regions, suggesting the SCN is important for promoting flexible semantic cognition (*Jefferies et al., 2020*). In this way, our results extended prior observations from univariate analyses by revealing that more complex representations and/or processes were implemented in transmodal areas during more controlled patterns of retrieval. Although regions towards the DMN apex of the principal gradient were associated with lower dimensionality of neural representation, we also observed higher sensitivity to the manipulation of associative strength in these areas. This suggests that transmodal regions adopt low-dimensional states when retrieval is adequately constrained by semantic long-term memory and higher-dimensional states when more idiosyncratic patterns of retrieval are generated.

It is worth noting that not all brain regions showed the expected pattern in the dimensionality analysis – especially when considering the global dimensionality of all semantic trials, as opposed to the influence of strength of association in the semantic task. In particular, the limbic network, including regions of ventral ATL thought to support a heteromodal semantic hub, showed significantly higher dimensionality than sensory-motor areas – these higher-order regions are expected to show lower dimensionality corresponding to more abstract representations. However, this analysis does not assess the psychological significance of data dimensionality differences (unlike our contrast of strong and weak associations, which are more interpretable in terms of semantic cognition). Limbic regions are subject to severe distortion and signal loss in functional MRI, which might strongly influence this metric. Future studies using data acquisition and analysis techniques that are less susceptible to this problem are required to fully characterise global dimensionality and its relation to the principal gradient.

In conclusion, we found that when strong associations were retrieved, and therefore cognition was well aligned with semantic information in long-term memory, the principal gradient was strengthened, suggesting greater separation of heteromodal and unimodal connectivity. When less related concepts were linked, this dimension of connectivity was reduced in strength as semantic control regions near the apex of the gradient separated from DMN to generate more flexible and original responses. In addition, semantic regions near the apex of the principal gradient showed higher dimensionality for weak associations, with activation patterns in these regions predicting individual differences in the similarity of semantic ratings. These results reveal that the principal gradient provides an organising principle for semantic cognition in the cerebral cortex.

## Methods
### Participants

A group of 36 healthy participants aged 19–35 years (mean age = 21.97 ± 3.47 years; 19 females) was recruited from the University of York. They were all right-handed, native English speakers, with normal or corrected-to-normal vision and no history of psychiatric or neurological illness. The study

was approved by the Research Ethics Committee of the York Neuroimaging Centre (Project number: P1391). All volunteers provided informed written consent and received monetary compensation or course credit for their participation. Five participants were not included in the data analysis: one participant had poor behavioural performance (no link made on 32% of trials), another withdrew during scanning, one scan showed a structural anomaly, one had missing volumes and another showed excessive head movement (movement >1.55 mm). Of the 31 participants included in the analysis, none showed (absolute) movement greater 1.2 mm. This study provides a novel analysis of a dataset first reported by *Krieger-Redwood et al., 2022*.

## Task materials and procedure

The stimuli were 144 English concrete noun pairs. Because abstract and concrete meanings, and taxonomic and thematic semantic relations, at least partially recruit different brain mechanisms (*Mkrtychian et al., 2019*; *Mirman et al., 2017*; *Montefinese, 2019*; *Wang et al., 2010*; *Schwartz et al., 2011*), we excluded abstract nouns and pairs of items drawn from the same taxonomic category, so that only thematic links were evaluated. We manipulated semantic association strength between the pairs of words using word2vec, with word-pairs ranging from completely unrelated (minimum word2vec = −0.05) to highly related (maximum word2vec = 0.72). The association strength did not show significant correlation with word frequency ($r = −0.010$, $p = 0.392$), concreteness ($r = −0.092$, $p = 0.285$), or imageability ($r = 0.074$, $p = 0.377$). Participants were viewed these word-pairs on screen, for 4.5 s, and were asked to identify a link between the items during this time. Participants rated the strength of the link they formed after each trial on a 5-point scale (0, no link; 1–4, weak to strong).

We used a slow event-related design, with trials lasting 13.5 s. Each trial began with a visually presented word-pair for 4.5 s; during this period, participants were tasked with identifying a link between the words. Following the link generation for each trial, there was 1.5 s to rate the strength of the link they retrieved. Semantic trials were separated by an easy chevron task for 6 s: participants pressed buttons to indicate whether each chevron faced left or right, with 10 chevrons presented. Finally, there was 1.5 s of fixation to alert the participant to the upcoming trial. Participants finished three functional runs, each lasting 11 min and 45 s. Immediately following the scanning session, participants were asked to recall and describe the link that they formed in the scanner. Using participants' post-scan recall of the links that they formed, we analysed the uniqueness of each response. These values were expressed as a proportion of the total sample who gave that particular response, ranging from.03 (a minimum of 1/31 participants) to 1 (a maximum of 31/31 participants).

## fMRI acquisition

Whole-brain structural and functional MRI data were acquired using a 3T Siemens Prisma MRI scanner utilising a 64-channel head coil, tuned to 123 MHz at the York Neuroimaging Centre, University of York. Three whole-brain functional runs were acquired using a multi-band multi-echo (MBME) EPI sequence, each lasting for 11.45 min (TR = 1.5; TEs = 12, 24.83, and 37.66 ms; flip angle $\theta = 75°$; FoV = 240 × 240 mm, resolution matrix = 80 × 80, and slice thickness = 3 mm; 455 volumes per run; GRAPPA acceleration factor = 3; multi-band acceleration factor = 2). Structural T1-weighted images were acquired using an MPRAGE sequence (TR = 2.3 s, TE = 2.26 s; flip angle = 8; FOV = 256 × 256 mm, matrix = 256 × 256, and slice thickness = 1 mm). A total of 176 sagittal slices were acquired to provide high-resolution structural images of the whole brain.

## fMRI data analysis

### Multi-echo data pre-processing

This study used an MBME scanning sequence. We used TEDANA to combine the images (*DuPre et al., 2021*; *Kundu et al., 2013*; *Kundu et al., 2012*). Before images were combined, some pre-processing was performed. FSL_anat (https://fsl.fmrib.ox.ac.uk/fsl/fslwiki/fsl_anat) was used to process the anatomical images, including re-orientation to standard (MNI) space (fslreorient2std), automatic cropping (robustfov), bias-field correction (RF/B1 – inhomogeneity-correction, using FAST), linear and non-linear registration to standard-space (using FLIRT and FNIRT), brain extraction (using FNIRT, BET), tissue-type segmentation (using FAST), and subcortical structure segmentation (FAST). The multi-echo data were pre-processed using AFNI (https://afni.nimh.nih.gov/), including de-spiking (3dDespike), slice timing correction (3dTshift), and motion correction (3daxialise, deoblique) of all

echoes aligned to the first echo (with a cubic interpolation). The script used to implement the pre-processing TEDANA pipeline is available at OSF (https://osf.io/mkgcy/). The TEDANA outputs (dn_ts_oc) in native space were filtered in the temporal domain using a non-linear high-pass filter with a 100-s cut-off. A two-step registration procedure was used whereby EPI images were first registered to the MPRAGE structural image (*Jenkinson and Smith, 2001*). Registration from MPRAGE structural image to standard space was further refined using FNIRT non-linear registration (*Anderson et al., 2007*). Functional images were resampled to 3-mm isotropic voxels for all analyses. Because the BOLD signal reaches its peak around 4–8 s after the stimuli's onset, the brain data were averaged within-trial across three time points for the semantic task (4.5–9 s after trial onset; TRs 4–6). To avoid potential contamination of neural responses from task-switching processes and the forthcoming semantic trial, we selected two time points for the chevron task (12–15 s after trial onset; TRs 9–10), resulting in one pattern of brain activity for the semantic and chevron tasks per trial. All analyses were performed in standard volume space.

## Network definition

The networks were taken from previous meta-analytic studies of the SCN and MDN (*Jackson, 2021*; *Fedorenko et al., 2013*). Within these networks, we selected (1) semantic control specific areas, which did not overlap with MDN; (2) multiple-demand specific regions, which did not overlap with SCN; (3) shared control regions, identified from the overlap between MDN and SCN; and (4) semantic regions not implicated in control; these were identified using Neurosynth (search term 'semantic'; 1031 contributing studies; http://www.neurosynth.org/analyses/terms/), removing regions that overlapped with the two control networks to identify regions associated with semantic representation or more automatic aspects of semantic retrieval, mostly within DMN (e.g. in lateral temporal cortex and AG).

## Cortical gradient analysis

Cortical gradient analysis was performed using The BrainSpace Toolbox (https://brainspace.readthe-docs.io/). Due to limitation of computational resources, we chose to use the default Schafer's 400 region parcellation in Brainspace. We performed gradient decomposition on a group-averaged parcel-by-parcel matrix of informational connectivity, rather than extracting the simple Pearson time-series correlation between pairs of regions, since informational connectivity corresponds to the similarity in the multivariate response of pairs of brain regions at different points in time (reflecting the multivariate response to trials in this experiment). In this way, informational connectivity is thought to reflect similarity in the representational states of parcels during a task.

We grouped the semantic judgements into strongly associated (top 1/3 trials, with a mean rating strength of 3.51) and weakly associated (bottom 1/3 trials, with a mean rating strength of 1.59) sets, according to the strength of association between the words within each trial (as measured by word2vec, widely used in the field, *Kivisaari et al., 2019*; *Pereira et al., 2016*). Between-trial pattern similarity matrices were produced for each condition. All within-run pairs of trials were excluded from the analysis to reduce the influence of temporal autocorrelation on pattern similarity. Next, the informational connectivity matrix was constructed by correlating the between-trial pattern similarity matrix between pairs of parcels for each condition. Lastly, the BrainSpace Toolbox was applied to the informational connectivity matrix to estimate the principal gradient, with the following parameters: kernel function: cosine similarity; dimensionality reduction: PCA.

For each trial, we also extracted the TRs associated with the chevron task. These judgements were again divided into two sets, based on the strength of the word association that preceded them. This allowed us to conduct a control analysis, examining the semantic manipulation at a time point when it should not affect the gradient decomposition results in a systematic fashion.

To confirm the advantage of the informational connectivity method, a supplementary analysis was conducted using a more standard measure of functional connectivity (i.e. Pearson correlation of averaged time series between any pair of ROIs). Gradient components extracted in this fashion proved to be insensitive to the effects of associative strength. These results are provided in Supplementary Materials (*Figure 2—figure supplement 5*).

## Second-order representational similarity analysis

A semantic model capturing the expected similarity in semantic processing across individuals was constructed by measuring the correlation of their ratings of associative strength across trials. For the brain data, we then calculated the between-trial pattern similarity for each participant using a searchlight approach: for each voxel, signals were extracted from a cube region containing 125 surrounding voxels (15 × 15 × 15 mm). A brain model capturing the neural similarity between each pair of participants was constructed for each voxel in MNI space by calculating the correlation of representational similarity patterns at that location. A second-order representational similarity analysis was then conducted by correlating the semantic and brain models to assess brain-behaviour alignment at the group level. All *r* values were Fisher *z* transformed before carrying them forwards into next-step analyses. To reduce the effect of temporal autocorrelation on pattern similarity, within-run pairs of trials were excluded from the analyses.

## Estimating the dimensionality of representational space

Brain regions that show a more diverse pattern of response across their voxels across trials and/ or participants are expected to contribute in a more complex or varied fashion to a task (*Diedrichsen et al., 2013*; *Rigotti et al., 2013*). To estimate the dimensionality of the representational space supporting the semantic processing of each word-pair in each parcel, we concatenated brain activation maps across participants into a four-dimensional data matrix for each trial, then applied PCA using a searchlight approach as above to measure the voxels' heterogeneity locally. The input matrix contained 31 ×125 features (31 participants by 125 voxels in each 5 × 5 × 5 searchlight cube). PCA is a linear approach that transforms the data to a low-dimensional space represented by a set of orthogonal components that explain maximal variance. We measured the dimensionality of each cube by calculating how many components were needed to explain more than 90% of the variance, reflecting the voxels' functional heterogeneity in each local area. To check the robustness of our findings, we also assessed dimensionality by calculating how many components were required to explain more than 60% and 75% of the variance. Next, we examined the effect of strength of association on the dimensionality of the neural response (i.e. between strongly and weakly associated trials) in the semantic task, across the whole group of participants. As a control analysis, we also examined dimensionality differences between strongly and weakly associated trials in the chevron task as a function of the associative strength of the semantic trial that preceded it. These results are provided in Supplementary Materials.

## Acknowledgements

This work was sponsored by the European Research Council (Project ID: 771863 - FLEXSEM), MRC Career Development Award MR/V031481/1, and The Rosetrees Trust A1699.

## Additional information

### Funding

| Funder | Grant reference number | Author |
| --- | --- | --- |
| European Research Council | 771863 - FLEXSEM | Elizabeth Jefferies |
| Medical Research Council | MRC Career Development Award MR/V031481/1 | Ajay Halai |
| Rosetrees Trust | A1699 | Ajay Halai |

The funders had no role in study design, data collection, and interpretation, or the decision to submit the work for publication.

### Author contributions

Zhiyao Gao, Conceptualization, Resources, Data curation, Software, Formal analysis, Validation, Investigation, Visualization, Methodology, Writing - original draft, Project administration, Writing - review

and editing; Li Zheng, Formal analysis, Methodology, Writing - review and editing; Katya Krieger-Redwood, Data curation, Investigation, Writing - review and editing; Ajay Halai, Methodology, Writing - review and editing; Daniel S Margulies, Jonathan Smallwood, Writing - review and editing; Elizabeth Jefferies, Conceptualization, Supervision, Funding acquisition, Writing - original draft, Writing - review and editing

## Author ORCIDs

Zhiyao Gao http://orcid.org/0000-0002-8909-8096

## Ethics

The study was approved by the Research Ethics Committee of the York Neuroimaging Centre (Project number: P1391). All volunteers provided informed written consent and received monetary compensation or course credit for their participation.

## Decision letter and Author response

Decision letter https://doi.org/10.7554/eLife.80368.sa1
Author response https://doi.org/10.7554/eLife.80368.sa2

---

# Additional files

## Supplementary files

• Transparent reporting form

• Supplementary file 1. Spin correlation between gradients estimated by traditional connectivity using resting state data and informational connectivity using semantic and chevron neural responses in each hemisphere.

## Data availability

Experiment materials, behavioural data, source data for producing the figures, brain parcellation template, and group-level neuroimaging data (gradient-relevant analysis) are accessible in the Open Science Framework at https://osf.io/mkgcy/. The Neurovault collection provides the processed version of the dataset for the other analyses, including neural dimensionality and second-order representational analysis: https://neurovault.org/collections/12539/. All analysis codes and software used in this study have been uploaded onto osf: https://osf.io/mkgcy/, which include but are not limited to the gradient analysis (Matlab), dimensionality analysis (python), and second-order RSA analysis (Matlab). The conditions of our ethical approval do not permit public archiving of the data because participants did not provide sufficient consent for the release of their biomedical data. Researchers who wish to access the data should contact the Research Ethics and Governance Committee of the York Neuroimaging Centre, University of York, or the corresponding authors. Data will be released to researchers when this is possible under the terms of the GDPR (General Data Protection Regulation). The decision as to whether the data can be reused and how access can be provided will be taken by the Research Ethics and Governance Committee of the York Neuroimaging Centre; data access arrangements are likely to exclude commercial use of the data.

The following datasets were generated:

| Author(s) | Year | Dataset title | Dataset URL | Database and Identifier |
|---|---|---|---|---|
| Gao Z | 2021 | Flexing the principal gradient of the cerebral cortex to suit changing semantic task demands | https://osf.io/mkgcy/ | Open Science Framework, mkgcy |
| Gao Z | 2022 | Flexing the principal gradient of the cerebral cortex to suit changing semantic task demands | https://identifiers.org/ neurovault.collection: 12539 | NeuroVault, collection:12539 |

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
