## [Editor Report]

This work provides important new insights into how semantic association strength influences the function and relationships across brain regions along a topographical structure of cerebral cortex. A principal gradient with the separation of default mode network from sensory-motor systems represents a hallmark of the retrieval of strong conceptual links. This study will be of interest to cognitive neuroscientists, especially those who are interested in semantic cognition.

---

## [Decision Letter]

**Decision letter after peer review:**

Thank you for submitting your article "Flexing the principal gradient of the cerebral cortex to suit changing semantic task demands" for consideration by *eLife*. Your article has been reviewed by 3 peer reviewers, and the evaluation has been overseen by a Reviewing Editor and Chris Baker as the Senior Editor. The following individuals involved in the review of your submission have agreed to reveal their identity: Xi-Nian Zuo (Reviewer #1); Wei Wu (Reviewer #3).

Essential revisions:

1. The level of analysis is not explained. For instance, for the cortical gradient analysis: are the gradients computed subject by subject and then averaged? Or is it the parcel-by-parcel matrices that are averaged? Or is it a fixed-effect analysis without considering the participant random factor? There are no error bars in Figure S1, so I assume the PCA is performed on the group average. It should be clearly stated and explained, not only for the gradient analysis but also for the dimensionality analysis and the RSA.

2. The performed statistical tests are not sufficiently explained. For example, what are the confidence intervals of their results? What is Pspin? What statistical test was performed to confirm the mediation analysis (if any)? What are the error bars in Figures 2 and 3?

3. Justification is needed for the selection of the time window for the analyses (TRs 4-6 in the semantic task and 9-10 in the chevron task). The authors indicate that the semantic trials are separated by an easy chevron task for 6 s (Line 619), but in Figure 1A this control task lasts for 4.5 s, which is a bit confusing. Also, the authors suggest 10 chevrons were presented one after another in each trial (Line 621). I wonder if 6 s was long enough for participants to look at so many chevrons and make judgements.

4. I wonder if the authors could provide more explanation about why they chose to exclude abstract nouns and pairs of items drawn from the same taxonomic category (Line 609).

5. How do response times vary according to the strength of association? How are response times taken into account in the analysis? For example, if trials with low association strength are longer, how is it taken into account?

6. How was the sample size (N=36) determined? Please clarify. Please provide statistical power, effect size, confidence interval, and corrections if multiple comparisons done for all the reports of statistics.

7. As a major manipulation in the experiment, it is not very clear how the authors split/define their stimuli into strong and weak semantic association conditions. If I understood correctly, word2vec was used to measure the association strength in each pair of words. Then the authors grouped the top 1/3 association strength trials as a "strong association" condition and the bottom 1/3 as "weak association" (Line 689), and all analyses comparing the effect of "strong vs. weak association" were conducted with data from these two subsets of stimuli. However, in multiple places, the authors indicate the association strength of their stimuli ranges from completely unrelated to weakly related to highly related (Line 612, Line 147, Line 690, and the examples in Figure 1B). This makes me wonder if the trials with bottom 1/3 association strength (i.e., those were used in the current study) are actually "unrelated/no association" trials (more like a baseline condition), instead of "weak association" trials as the authors claimed. These two situations could be different regarding how they engage semantic knowledge and control processing. Besides, I am very interested in what will the authors find if they compare all three conditions (i.e., unrelated vs. weak association vs. strong association).

8. Because the comparison between weak vs. strong association conditions is the key to the current study, I feel it might be better to introduce more about the stimuli in these two conditions. Specifically, the authors only suggested the word pairs fell in these two conditions varied in their association strength, but how about other psycholinguistic properties that could potentially confound their manipulation? For example, words with higher frequency and concreteness may engage more automatic/richer long-term semantic information and words with lower frequency and concreteness need more semantic control. I feel there may be a possibility that the effect of semantic association was partly driven by the differences in these measures in different conditions.

9. Behavior: Correlating the mean response and word2vec distance is not very informative (except for confirming that word2vec distance is meaningful for the mean population). Is this correlation accurate at the group level? i.e., if we compute the correlation coefficient per individual and then test the mean of the coefficient at the group level, is it significant?

10. I wonder if the uniqueness of response (Line 627) could be used to calculate a more straightforward measure of association strength for word pairs than word2vec. That is, for each trial, the more participants link the two words in the same way, the stronger the link is. And the fewer participants link the words in the same way, the weaker and more sparse the association is.

11. Some correlation coefficients are puzzlingly high (e.g., 0.84 in Figure 1C). Please discuss.

12. One strength of the gradient method is its continuous spatial variation in regard to the brain divisions, thus a spatially continuous brain parcellation. Thus, it would be more taking such advantage by using a voxel/vertex analysis than the 400-parcel analysis. What is the tenet for the current work to use the large-parcel gradient analysis? Is this something due to the limitation of the computational resource? If so, please clarify, and if not, please discuss.

13. It seems different cortical surface models are used for rendering the spatial maps (e.g., Figure 2ABC vs Figure 2E vs Figure 3A). Please use an identical surface model for better direct comparisons across different visualizations.

14. It might be good practice to add labels (e.g., t value, r value etc.) next to the colour bars in the activation maps, which could make the figures easier to understand.

15. The dimensionality analysis in the current study is novel and interesting. In this section, the authors linked decreasing dimensionality with more abstract and less variable representations. However, most results here were built based on the comparison between the dimensionality effects for strong and weak association conditions. I wonder if these conclusions can be generalised to results within each condition and across different regions (i.e., regions having lower dimensionality are doing more abstract and cross-modal processing). If so, I am curious why the ATL (a semantic "hub") in Figure 3A has higher dimensionality than the sensory-motor cortices (quite experiences related) and AG (another semantic "hub").

16. A more extensive discussion should be accompanied by the revision to highlight the implications and values of the present work for brain-mind development from a lifespan perspective.

17. Why do the authors use a binary separation of the distance while it is a continuous measure? If this is an actual modulation, the intermediary bin (intermediate values not included in the analyses) should present an intermediary profile.

---

## [Author Response]

Essential revisions:1. The level of analysis is not explained. For instance, for the cortical gradient analysis: are the gradients computed subject by subject and then averaged? Or is it the parcel-by-parcel matrices that are averaged? Or is it a fixed-effect analysis without considering the participant random factor? There are no error bars in Figure S1, so I assume the PCA is performed on the group average. It should be clearly stated and explained, not only for the gradient analysis but also for the dimensionality analysis and the RSA.

Sorry that we did not make these details clear in our initial submission. We now include more details about the level of analysis in both the method and Results sections.

a. The cortical gradients were computed by using group averaged parcel-by-parcel matrices. Please see page 35:

‘We performed gradient decomposition on a group-averaged parcel-by-parcel matrix of informational connectivity, rather than extracting the simple Pearson time series correlation between pairs of regions, since informational connectivity corresponds to the similarity in the multivariate response of pairs of brain regions at different points in time (reflecting the multivariate response to trials in this experiment).’

Also see page 10:

‘PCA was used to decompose the group-averaged connectivity matrices, with the resulting components (or gradients) reflecting spatial patterns of connectivity across the cortex.’

b. We concatenated all participants’ neuroimaging data into one 4D file for one single trial and performed PCA on this to estimate the dimensionality map, then iterated the same procedure for all trials. The statistical test was conducted to test the association strength effect between conditions. In this way, the analysis considers group-level effects. Please see page 37.

‘Next, we examined the effect of strength of association on the dimensionality of the neural response (i.e., between strongly and weakly associated trials) in the semantic task, across the whole group of participants.’

c. The RSA analysis was conducted on the group level. We firstly constructed neural pattern similarity matrices between trials at the individual level, then correlated these matrices between participants using a searchlight approach. We also calculated the similarity of behavioral rating patterns between participants to construct a semantic rating similarity matrix. Lastly, we correlated these two group level matrices between neural and semantic levels. Please see page 36.

‘A second-order representational similarity analysis was then conducted by correlating the semantic and brain models to assess brain-behaviour alignment at the group level.’

2. The performed statistical tests are not sufficiently explained. For example, what are the confidence intervals of their results? What is Pspin? What statistical test was performed to confirm the mediation analysis (if any)? What are the error bars in Figures 2 and 3?

Thanks very much for identifying these points where more information is needed. We have significantly updated the statistical reporting throughout the manuscript according to the reviewer’s comments.

a. Confidence intervals (95%) were used and are now systematically reported in the manuscript. Please see page 13:

‘Simple t-tests revealed significant differences in principal gradient values for the semantic task across SCN and SCN+MDN compared with MDN regions (t(30) = 3.259, p = 0.0167, 95% CI = [0.279 ~ 1.215], Cohen’s d = 0.827; and t(30) = 3.904, p = 0.003, 95% CI = [0.334 ~ 1.066], Cohen’s d = 0.965, respectively).’

And see page 22:

‘Consistent with our prediction, we found the correlation between the principal gradient and semantic-brain alignment could be partially explained by this dimensionality difference (indirect effect: a x b = 0.0053, 95% CI = [0.0039 ~ 0.01], p < 0.001, explaining 51.7% variance; direct effect: c’ = 0.0049, 95% CI = [0.0026 ~ 0.01], p < 0.001).’

b. The error bars in Figure 2D and Figure 3C represent the standard error of gradient differences and of dimensionality differences across parcels in each brain network.

Please see page 20:

‘Error bars represent the standard error of dimensionality differences across parcels within each network.’

c. Pspin represents the significance of the spin permutation test. The significance of the correlation between gradients estimated by our task-based data and gradients estimated by using resting state data were assessed by using spin permutation to control the spatial autocorrelation in MRI data. Please see page 10:

‘This similarity between cortical gradients was established by using spin permutation tests, which preserve the spatial autocorrelation present within gradients. Statistical significance was determined by comparing with a null distribution using the permutation approach (5000 permutations), in which we randomly rotated the spatial patterns and measured the correlations between maps that preserve spatial autocorrelation (Alexander-Bloch et al., 2018). The first gradient, accounting for the greatest variance in informational connectivity, was highly overlapping with the principal gradient of intrinsic connectivity identified by prior work (Margulies et al., 2016), for both the semantic and chevron tasks (Spin permutation p value: P_spin_ < 0.001), see Figure 2 —figure supplement 1.’

d. The significance of mediation effect was tested by using a bootstrapping approach which was highly recommended recently (Shrout and Bolger, 2002).

Please see page 22:

‘To test this hypothesis, we performed a mediation analysis in which the principal gradient was the predictor, semantic-brain alignment was the dependent variable, and the effect of strength of association on the dimensionality of the neural response was the mediator. The significance of the mediation effect was tested by using the bootstrapping approach with 5000 samples. Consistent with our prediction, we found the correlation between the principal gradient and semantic-brain alignment could be partially explained by this dimensionality difference (indirect effect: a x b = 0.0053, 95% CI = [0.0039 ~ 0.01], p < 0.001, explaining 51.7% variance; direct effect: c’ = 0.0049, 95% CI = [0.0026 ~ 0.01], p < 0.001).’

3. Justification is needed for the selection of the time window for the analyses (TRs 4-6 in the semantic task and 9-10 in the chevron task). The authors indicate that the semantic trials are separated by an easy chevron task for 6 s (Line 619), but in Figure 1A this control task lasts for 4.5 s, which is a bit confusing. Also, the authors suggest 10 chevrons were presented one after another in each trial (Line 621). I wonder if 6 s was long enough for participants to look at so many chevrons and make judgements.

a. Thanks for pointing this out. The BOLD signal reaches its peak around 4 ~ 8s after the stimuli’s onset. Considering the TR is 1.5s, we chose TRs (4 ~ 6) which correspond to 4.5 ~ 9s after stimuli’s onset to represent the semantic neural responses: this is the period expected to correspond most strongly to the semantic response. The chevron task started at the 5^th^ TR after stimuli onset but considering task-switching effects may contaminate the first few chevron trials, and to avoid potential contamination of neural responses from the forthcoming semantic period, we excluded the first chevron TR (started at TR 9) and ended this period one TR earlier (at TR 10); this period covers the peak response period for the chevron task 6 ~ 9s after trial onset. Please see page 34:

‘Because the BOLD signal reaches its peak around 4 ~ 8s after the stimuli’s onset, the brain data were averaged within-trial across three time points for the semantic task (4.5~9s after trial onset; TRs 4-6). To avoid potential contamination of neural responses from task-switching processes and the forthcoming semantic trial, we selected two time points for the chevron task (12~15s after trial onset; TRs 9-10), resulting in one pattern of brain activity for the semantic and chevron tasks per trial.’

b. The chevron task lasted for 6s in each trial. There was a typo in Figure 1A, which has been corrected. Sorry for the confusion caused by this. Please see page 9:

c. The participants’ averaged reaction time was 0.336s which was lower than the 0.6s that participants had to perform each chevron decision. The accuracy was 72.6%. Therefore, the chevron task was challenging enough to engage the participant, but not so challenging that they could not do it. Please see page 8.

‘On the chevron task, the average reaction time was 0.336s and the accuracy was 72.6%; these data suggest the task was challenging enough to engage the participant but not so difficult that they could not do it. The fast pace of this task reduced participants’ ability to think about semantic links across trials.’

4. I wonder if the authors could provide more explanation about why they chose to exclude abstract nouns and pairs of items drawn from the same taxonomic category (Line 609).

There is a debate about the nature of abstract and concrete concepts, with previous studies suggesting that they recruit partially different brain mechanisms (Montefinese, 2019, Wang et al., 2010). Previous studies have also suggested that there are distinct differences between the processes supporting taxonomic and thematic knowledge: taxonomic knowledge is based on shared features (e.g., dog – cat), while thematic knowledge is based on co-occurrence in events or scenarios (Mirman et al., 2017). Again, studies have identified some differences in the neural response to these different types of semantic relationships (Mirman et al., 2017, Schwartz et al., 2011, Lewis et al., 2015). To avoid potential confounds caused by these differences, we excluded taxonomic noun pairs in the current study. We have now provided an explanation for this decision in the method of our manuscript. Please see page 31.

‘Because abstract and concrete meanings, and taxonomic and thematic semantic relations, at least partially recruit different brain mechanisms (Mkrtychian et al., 2019, Mirman et al., 2017, Montefinese, 2019, Wang et al., 2010, Schwartz et al., 2011), we excluded abstract nouns and pairs of items drawn from the same taxonomic category, so that only thematic links were evaluated.’

5. How do response times vary according to the strength of association? How are response times taken into account in the analysis? For example, if trials with low association strength are longer, how is it taken into account?

We agree with the reviewer that response times vary according to the strength of association; it usually takes a longer time to verify semantic links for trials that are weakly associated (Teige et al., 2019, Gao et al., 2021). When participants make easier and harder semantic decisions, they are expected to spend more time on the harder task, since after a response has been made, participants are often not thinking about the task. However, our task was different, since participants had to generate a meaningful link between all pairs of words, irrespective of strength of association – they could do this for strongly-related items but also for unrelated items. They had 4.5 seconds to think about the relationship between the words and then rated the strength of the link that they had generated. In this way, the task structure does not give rise to clear differences in time on task across trials, although we acknowledge that there might be differences in the time spent on the initial generation of ideas compared with their elaboration.

We selected TR 4-6 to represent semantic signals in the brain since this corresponded to 6 ~ 9s after stimuli onset, which covers the peak signal of BOLD response (4 ~ 8s), please see more details from our reply to the comment 3.

Please see page 32:

‘We used a slow event-related design, with trials lasting 13.5s. Each trial began with a visually presented word-pair for 4.5s; during this period, participants were tasked with identifying and encouraged to keep thinking about a link between the words. Following the link generation for each trial, there was 1.5s to rate the strength of the link they retrieved.’

6. How was the sample size (N=36) determined? Please clarify. Please provide statistical power, effect size, confidence interval, and corrections if multiple comparisons done for all the reports of statistics.

No explicit power analysis was performed to determine the sample size in this experiment, however, the present sample size (N=31) is comparable to, or significantly exceeds, that of prior task-based fMRI studies examining the neural basis of semantic cognition (e.g., Ubaldi et al., 2022, J.Neuroscience; Gao et al., 2022, Cerebral Cortex; Sormaz et al., 2018, PNAS).

We have extensively modified the manuscript to add reports of statistical power, effect sizes, confidence intervals and we have explicitly stated whether corrections are performed for multiple comparisons, with this correction applied whenever relevant.

Please see page 13:

‘Simple t-tests revealed significant differences in principal gradient values for the semantic task across SCN and SCN+MDN compared with MDN regions (t(30) = 3.259, p = 0.0167, 95% CI = [0.279 ~ 1.215], Cohen’s d = 0.827; and t(30) = 3.904, p = 0.003, 95% CI = [0.334 ~ 1.066], Cohen’s d = 0.965, respectively).’

‘We found a significant interaction between brain network and task (F(2.53, 75.887) = 8.345, p < 0.001, η2p = 0.218 ) and significant main effects of task (F(1, 30) = 21.947, p < 0.001, η2p = 0.414) and network (F(2.01, 60.288) = 3.338, p = 0.042, η2p = 0.100), see Figure 2D.’

And see page 16:

‘The difference in principal gradient values between strong and weak association trials showed a significant interaction effect between functional networks and tasks (F(2.53, 75.887) = 8.345, p < 0.001, η2p = 0.218 ). Error bars represent the standard error of gradient difference across parcels within each network.’

And see page 22:

‘Consistent with our prediction, we found the correlation between the principal gradient and semantic-brain alignment could be partially explained by this dimensionality difference (indirect effect: a x b = 0.0053, 95% CI = [0.0039 ~ 0.01], p < 0.001, explaining 51.7% variance; direct effect: c’ = 0.0049, 95% CI = [0.0026 ~ 0.01], p < 0.001).’

7. As a major manipulation in the experiment, it is not very clear how the authors split/define their stimuli into strong and weak semantic association conditions. If I understood correctly, word2vec was used to measure the association strength in each pair of words. Then the authors grouped the top 1/3 association strength trials as a "strong association" condition and the bottom 1/3 as "weak association" (Line 689), and all analyses comparing the effect of "strong vs. weak association" were conducted with data from these two subsets of stimuli. However, in multiple places, the authors indicate the association strength of their stimuli ranges from completely unrelated to weakly related to highly related (Line 612, Line 147, Line 690, and the examples in Figure 1B). This makes me wonder if the trials with bottom 1/3 association strength (i.e., those were used in the current study) are actually "unrelated/no association" trials (more like a baseline condition), instead of "weak association" trials as the authors claimed. These two situations could be different regarding how they engage semantic knowledge and control processing. Besides, I am very interested in what will the authors find if they compare all three conditions (i.e., unrelated vs. weak association vs. strong association).

We appreciate the reviewer pointing this out. The association strength of word pairs ranged from completely unrelated to highly related, reflected in both word2vec values (-0.05 to 0.72) and participants’ semantic ratings (from 0 ~ 4). In this way, the reviewer is right that the bottom third of trials largely consisted of words that might be thought of as ‘unrelated’ in other contexts, although the task still required participants to come up with a meaningful way of linking these items – and in this way, we did not include any “no association” trials.

When we divided our stimuli into three sets as requested by the reviewer, the average semantic rating ranged from 1.59 (bottom 1/3) through 2.82 (middle 1/3) to 3.51 (top 1/3), with extremely significant difference between all sets of items (all raw p values were lower than 2.83e-14). Therefore, we have rephrased our description, now referring to the top and bottom 1/3 stimuli as more strongly and less strongly associated.

Please see page 35:

‘We grouped the semantic judgements into strongly associated (top 1/3 trials, with a mean rating strength of 3.51) and weakly associated (bottom 1/3 trials, with a mean rating strength of 1.59) sets, according to the strength of association between the words within each trial (as measured by word2vec, widely used in the field (Kivisaari et al., 2019, Pereira et al., 2016)).’

We have conducted additional analysis for the intermediary bin and compare it against the bottom for the gradient analysis and against the top 1/3 for the dimensionality analysis (compared to the baseline condition for each analysis), which did show a similar patten like the contrast between strong and weak association but with a smaller effect, thus representing an intermediary profile as expected. The correlation between the principle gradient difference between middle and weak association with the principle gradient value derived from resting state was also significant, see Figure 2 —figure supplement 3C, but its magnitude was smaller than what we reported in the main body of manuscript (r = 0.235 vs. r = 0.369). Given that the expected strongest effect is between top and bottom 1/3, thus, we have now included these results in the supplementary materials. Please see Figure 2 —figure supplement 3 in page 44.

8. Because the comparison between weak vs. strong association conditions is the key to the current study, I feel it might be better to introduce more about the stimuli in these two conditions. Specifically, the authors only suggested the word pairs fell in these two conditions varied in their association strength, but how about other psycholinguistic properties that could potentially confound their manipulation? For example, words with higher frequency and concreteness may engage more automatic/richer long-term semantic information and words with lower frequency and concreteness need more semantic control. I feel there may be a possibility that the effect of semantic association was partly driven by the differences in these measures in different conditions.

Thanks for raising this point. We have performed additional control analysis to examine the relationship between association strength and psycholinguistic features according to the reviewer’s suggestion. The association strength did not show significant correlation with word frequency (r = -0.010, p = 0.392), concreteness (r = -0.092, p = 0.285) or imageability (r = 0.074, p = 0.377). Direction comparison of these psycholinguistic features between more strongly and les strongly associated word-pairs also did not any significant difference: frequency (t = 0.912, p = 0.364), concreteness (t = 1.576, p = 0.119), imageability (t = 1.451, p = 0.153). Please see page 32:

‘The association strength did not show significant correlation with word frequency (r = -0.010, p = 0.392), concreteness (r = -0.092, p = 0.285) or imageability (r = 0.074, p = 0.377).’

9. Behavior: Correlating the mean response and word2vec distance is not very informative (except for confirming that word2vec distance is meaningful for the mean population). Is this correlation accurate at the group level? i.e., if we compute the correlation coefficient per individual and then test the mean of the coefficient at the group level, is it significant?

We agree with the reviewer that the correlation analysis conducted at the group level could not reveal the relationship for the individual level, though it does provide us clear and strong evidence that word2vec was a reliable measure of association strength. Following the reviewer’s suggestion, we performed a linear mixed effect model analysis to test whether word2vec was a reliable predictor of semantic ratings at the individual level. Participants were included as a random effect and word2vec was treated as a predictor of semantic ratings. We used the likelihood ratio test to compare models with and without word2vec to determine whether the inclusion of word2vec significantly improved the model fit. In line with the correlation analysis between mean response and word2vec, the results revealed a similar significant relationship (χ2(1) = 2266.2, p < 2.3-e16). Considering that our key conclusions nearly were all based on the group-level analysis, thus we think the group-level correlation analysis is an informative and suitable measure here.

10. I wonder if the uniqueness of response (Line 627) could be used to calculate a more straightforward measure of association strength for word pairs than word2vec. That is, for each trial, the more participants link the two words in the same way, the stronger the link is. And the fewer participants link the words in the same way, the weaker and more sparse the association is.

We agree with the reviewer that the uniqueness of responses is closely associated with association strength: fewer participants link the words in the same way when the association strength is weaker. Theoretically, with enough data, the uniqueness of responses could reflect association strength in a stable and precise way. However, the current sample was not large enough to fully reflect shared knowledge in the culture, unlike word2vec. Word2vec was trained on a super large corpus dataset which aims to measure the cooccurrence of words in similar scenarios; this not only captures the definition of association strength for thematic knowledge but has proved a reliable and excellent predictor of association strength in the field. This is why we used word2vec as the index of association strength in the current study.

11. Some correlation coefficients are puzzlingly high (e.g., 0.84 in Figure 1C). Please discuss.

Word2vec was trained on a large corpus dataset using a deep neural network, and other studies have already found it is an excellent predictor of association strength (Hoffman, 2018, Pereira et al., 2016): in this way, this strong correlation is not unexpected. In another study using similar design in our lab (Gao et al., 2021), we collected 192 word pairs and calculated the correlation coefficient between word2vec and semantic ratings, on average across the whole sample. We obtained similar results: Word2vec was strongly positively correlated with rating score (r = 0.773, p < 0.0001), showing that people were more likely to judge word pairs as related when they had high word2vec values.

12. One strength of the gradient method is its continuous spatial variation in regard to the brain divisions, thus a spatially continuous brain parcellation. Thus, it would be more taking such advantage by using a voxel/vertex analysis than the 400-parcel analysis. What is the tenet for the current work to use the large-parcel gradient analysis? Is this something due to the limitation of the computational resource? If so, please clarify, and if not, please discuss.

We agree with reviewer that it would be ideal to measure brain gradients using a voxel/vertex analysis. However, as the reviewer pointed out, the computational resource would be extremely high, as the time series of neural similarity matrices is far larger than traditional averaged univariate time series in each voxel/vertex-centered cubes (searchlight-like analysis). Moreover, 400-parcels is the default option within the BrainSpace package. We have clearly stated the reason for using this parcellation in the revised method, please see in page 35.

‘Due to limitation of computational resources, we chose to use the default Schafer’s 400 region parcellation in Brainspace.’

13. It seems different cortical surface models are used for rendering the spatial maps (e.g., Figure 2ABC vs Figure 2E vs Figure 3A). Please use an identical surface model for better direct comparisons across different visualizations.

Thank you very much for raising this. We have replotted all relevant figures according to the reviewer’s comment. Now all spatial maps are projected on the identical cortical surface model.

14. It might be good practice to add labels (e.g., t value, r value etc.) next to the colour bars in the activation maps, which could make the figures easier to understand.

Thanks for raising this, we have now updated all relevant figures according to the reviewer’s comment.

15. The dimensionality analysis in the current study is novel and interesting. In this section, the authors linked decreasing dimensionality with more abstract and less variable representations. However, most results here were built based on the comparison between the dimensionality effects for strong and weak association conditions. I wonder if these conclusions can be generalised to results within each condition and across different regions (i.e., regions having lower dimensionality are doing more abstract and cross-modal processing). If so, I am curious why the ATL (a semantic "hub") in Figure 3A has higher dimensionality than the sensory-motor cortices (quite experiences related) and AG (another semantic "hub").

The dimensionality and its relationship to the cortical gradient was also examined for each condition, indeed, we also assessed whether this relationship was influenced by associative strength, averaging estimates derived for sets of four trials with similar word2vec values using a ‘sliding window’ approach. There was a negative correlation between overall dimensionality (averaged across all trials) and principal gradient values. And the magnitude of this negative relationship increased as a function of the association strength (and importantly, all correlation values (in Figure 3D) are negative). So, we believe our conclusion could be generalized across conditions although it would be more difficult to observe this relationship if only a subset of trials was included in the analysis.

In our results, we observed higher dimensionality in ATL/frontal orbital cortex than sensory-motor cortices, which seems contradictory to our conclusion. However, these areas are subject to severe distortion and signal loss in functional MRI (Binney et al., 2010). This low tSNR may have produced a higher dimensionality estimation in PCA. We conducted a control analysis in which regions in the limbic network were removed due to their low tSNR, and the negative relationship between dimensionality and principal gradient remained unchanged (r = -0.346, p = 0.038).

Please see the Discussion on page 30.

‘It is worth noting that not all brain regions showed the expected pattern in the dimensionality analysis – especially when considering the global dimensionality of all semantic trials, as opposed to the influence of strength of association in the semantic task. In particular, the limbic network, including regions of ventral ATL thought to support a heteromodal semantic hub, showed significantly higher dimensionality than sensory-motor areas – these higher-order regions are expected to show lower dimensionality corresponding to more abstract representations. However, this analysis does not assess the psychological significance of data dimensionality differences (unlike our contrast of strong and weak associations, which are more interpretable in terms of semantic cognition). Limbic regions are subject to severe distortion and signal loss in functional MRI, which might strongly influence this metric. Future studies using data acquisition and analysis techniques that are less susceptible to this problem are required to fully characterize global dimensionality and its relation to the principal gradient.’

16. A more extensive discussion should be accompanied by the revision to highlight the implications and values of the present work for brain-mind development from a lifespan perspective.

Thanks for raising this important issue. We have included relevant discussion in the discussion part for this important question. Please see it on page 26.

‘The human capacity to retrieval meaning from sounds, visual objects, words and phrases far exceeds the capacities of other animal species, however, its development is poorly understood (Enge et al., 2021). One recent study (Dong et al., 2021) found that, in children, the overarching organizational gradient is anchored within the unimodal cortex, between somatosensory/motor and visual territories. While in adolescence, the principal gradient of connectivity transitions into an adult-like spatial framework, with the default network at the opposite end of a spectrum from primary sensory and motor regions. Whether the development of semantic system is paralleled with gradual transition of the principal gradient in the brain remain to be an open question and is required with future studies.’

17. Why do the authors use a binary separation of the distance while it is a continuous measure? If this is an actual modulation, the intermediary bin (intermediate values not included in the analyses) should present an intermediary profile.

Thanks very much for raising this point out. We have conducted additional analysis for the intermediary bin and compare it against the bottom for the gradient analysis and against the top 1/3 for the dimensionality analysis (compared to the baseline condition for each analysis), which did show a similar patten like the contrast between strong and weak association but with a smaller effect, thus representing an intermediary profile as expected. The correlation between the principle gradient difference between middle and weak association with the principle gradient value derived from resting state was also significant, see Figure 2 —figure supplement 3C, but its magnitude was smaller than what we reported in the main body of manuscript (r = 0.235 vs. r = 0.369). Given that the expected strongest effect is between top and bottom 1/3, thus, we have now included these results in the Figure 2 —figure supplement 3. Please see page 44.